# Potential Mechanisms Underlying Hypoxia-Induced Diabetes in a Rodent Model: Implications for COVID-19

**DOI:** 10.3390/children8121178

**Published:** 2021-12-14

**Authors:** Eung-Kwon Pae, Ronald M. Harper

**Affiliations:** 1School of Dentistry, University of Maryland, 650 W. Baltimore St., Baltimore, MD 21201, USA; 2Department of Neurobiology, University of California at Los Angeles, Los Angeles, CA 90095, USA; rharper@ucla.edu

**Keywords:** β-cell, chloride levels, COVID19-induced hypoxia, insulin, KCC2

## Abstract

Previous studies reported that repetitive hypoxia in rat pups reduces insulin secretion and elevates fasting blood glucose levels; these sequelae persisted for several months. This report describes how episodic hypoxic events elevate a chloride ion exporter, K^+^-Cl^−^ cotransporter-2 (KCC2), in the plasma membrane of insulin-secreting pancreatic β-cells. We assume that acute diabetic symptoms observed in rat pups with periodic oxygen desaturation could result from a lack of blood insulin levels due to disturbed β-cell function. This acute hypo-insulinemia may result from a disruption in chloride balance in β-cells arising from an imbalanced KCC2-NKCC1 (chloride exporter-importer) density as a consequence of periodic oxygen desaturation. Mechanistically, we postulate that a reduced insulin secretion due to the KCC2-NKCC1 imbalance subsequent to acute oxygen desaturation could result in hyperglycemia in rat pups, paralleling symptoms shown in patients with COVID-19 who experienced acute respiratory distress.

## 1. Introduction

Co-occurrence of acute respiratory distress and diabetes can be life-threatening. Brief repetitive hypoxic exposure to rats reduces insulin secretion and elevates fasting blood glucose levels; these sequelae continue for as long as 12 months [1]. Acute post-infection disruption of glycometabolic control commonly accompanies severe acute respiratory syndrome coronavirus 2 (SARS-CoV-2) infection [2,3]. We suspect that hypoxia, associated with acute respiratory distress, occurring after intrusion of the virus, underlies the development of ‘sudden-onset diabetes’ among people without a history of diabetes [3,4,5], rather than the levels of ‘cytokine storm’ measured in blood [6]. A cardinal sign of SARS-CoV-2 infection is oxygen desaturation <92%, although such oxygen desaturation levels may not be accompanied by dyspnea, depending on the patient [7]. Nevertheless, shortness of breath, in conjunction with an increased number of immature erythrocytes in circulation and a disrupted glycometabolic control are considered the most common sequelae of coronavirus disease 2019 (COVID-19). Although, currently, we have no direct evidence to link the hyperglycemia induced by intermittent hypoxic breathing and hypochloremia [8,9] found in some patients infected with the SARS-CoV-2, earlier studies [10,11] indicate that intermittent hypoxia (IH) exposure disrupts insulin secretion and raises the possibility of a mechanistic link for disturbed glucose regulation in COVID-19; we explored that possibility using our current in vivo and in vitro models.

Results from several rodent models of insulin homeostasis studies demonstrated that IH exacerbates pancreatic β-cell dysfunction [1,11]; however, tissue responses to hypoxia, in accord with insulin levels, appear to be tissue-/condition-specific [12]. Pancreatic β-cells sense and respond to secretagogues such as glucose, secreting insulin as the plasma membrane is depolarized. Previous studies have shown that the β-cell membrane depolarizes with an increase in chloride ion transit across the membrane [13,14]. In β-cells, [Cl^–^]_i_ is maintained above the assessed Nernstian value by net action of Cl^−^ importers [15]. Therefore, opening of Cl^−^ channels would result in efflux of chloride ions in β-cells. Increasing glucose levels sensed by the cell triggers membrane depolarization as high levels of intracellular Cl^−^ are maintained, resulting in insulin secretion [15,16]. Thus, maintaining intracellular chloride levels within an optimal range in this sequence is critical for normal glucose metabolism. Intracellular chloride ion concentration is regulated by the chloride importer Na^+^-K^+^-Cl^−^ cotransporter (NKCC1), and chloride exporter K^+^-Cl^−^ cotransporter (KCC2; a product of Slc12a5), which are found in various cells such as neuromuscular cells and pancreatic β-cells [17,18,19]. We suspect that this chloride-dependent mechanism reported in pancreatic β-cells operates independently from the well-known ATP-sensitive potassium channel (K_ATP_) mechanism of modulating insulin secretion as previously reported by others [13,15,16].

We observed that exposure of rat pups to brief IH, in which ambient oxygen levels alternate between 10% and 21%, results in decreased blood insulin levels accompanied by hyperglycemia and abnormal glucose tolerance 3 weeks post-challenge [1,20]. If these effects were not, however, associated with a reduction in the number of pancreatic β-cells or β-cell mass as exhibited previously [1,20], the low insulin levels in blood after IH exposure might be a consequence of β-cell dysfunction due to IH-mediated damage to the chloride homeostatic mechanism resulting in unbalanced chloride levels in β-cells.

We postulated that β-cells rely on NKCC1 and KCC2 transporters for controlling ingress and exit of chloride ions. If the intracellular chloride concentration in β-cells is decreased (due to an imbalance in the number of chloride importers/exporters), membrane depolarization would not occur; in which case, insulin secretion could decrease or fail. We proposed a mechanism rationalizing the cause–effect relationship between IH exposure and decreased serum insulin levels. The testable hypothesis is that an increase in KCC2 chloride exporters in the pancreatic β-cell membrane after IH-exposure would result in a reduction in intracellular chloride content, and lead to decreased insulin secretion. We exposed primary pancreatic islet cells and rat pups to IH, and quantified chloride transporters with respect to those under normal condition to assess effects on chloride balance.

## 2. Materials and Methods

### 2.1. Isolation of Islets and Subcellular Protein Fractionation

Pancreatic islets were harvested as previously described [20]. In brief, cold collagenase solution was injected into the pancreas through the common bile duct to separate pancreatic tissue from fat tissue immediate after euthanasia of male rat pups. The removed pancreas was digested with collagenase in a conical tube at 37 °C for 8 min, followed by washing twice using G-solution (1% BSA containing Hank’s balanced salt solution) to slow the digestive process. The tissue was filtered, centrifuged, and the pellet was re-suspended in Histopaque 1100 solution (1077 and 1119 mixed; Sigma-Aldrich, St. Louis, MO, USA) and separated by gradient centrifugation. The supernatant was taken and supplemented with 10% FBS and 1% Penicillin-Streptomycin mixture, and then was cultured at 37 °C 5% CO_2_ for 4 h.

### 2.2. In Vitro IH Treatment of Isolated Islets

An equal quantity of isolated islets in culture medium, harvested from multiple pups, were placed in a hypoxia chamber (Billups-Rothenberg Inc., San Diego, CA, USA), and chamber air was alternated between room air and a hypoxic gas mixture (1% O_2_, 5% CO_2_, 94% N_2_) every 10 min for ten cycles (3 h total). The treated and untreated-control cells were incubated at least 12 h before the next assays.

### 2.3. siRNA Mediated KCC2 Gene Knockdown and KCC2 Protein Inhibition

RNA interference methods were used to determine if regulation of transcription of KCC2 was responsible for insulin production and secretion. Control and IH-treated islets were transfected with siRNA targeting rat Slc12a5 (KCC2; Accession Number NM_134363) or with scrambled siRNA control (Sigma-Aldrich) using X-tremeGENE siRNA transfection reagent (Roche Diagnostics), and incubated for 2 days. The target sequences were: Slc12a5-2220, CAU UGU GGG CUC UGU CCU U; Slc12a5-2312, CUG AGA AGG UGA AGG GCU U. RNA interference effects were evaluated using real-time PCR assays of mRNA production.

To assess the effects of siRNA-mediated down-regulation, the amounts of insulin produced following siRNA-knockdown were compared to insulin amounts produced following pharmacologic inhibition of KCC2. To test for potential insulin release, equal amounts of harvested islets were cultured for 4 h in medium with 50 µM of KCC2 inhibitor [21] VU0240551 (N-(4-Methyl-2-thiazolyl)-2-[(6-phenyl-3-pyridazinyl) thio]-acetamide; Sigma-Aldrich), after which secreted insulin was quantified by ELISA.

### 2.4. ELISA

Assays for insulin protein quantification from blood or culture media were performed using an ELISA Kit (EMD Millipore Corp, Burlington, MA, USA), in accordance with the manufacturer’s protocol. Samples in an equal amount were incubated in insulin monoclonal antibody-coated plates, detected using a secondary biotinylated insulin antibody, and followed by streptavidin-conjugated horseradish peroxidase. Immobilized antibody–enzyme conjugates were quantified by monitoring color development in the presence of TMB (3,3′,5,5′-tetramethylbenzidine) substrate at 450 nm in a spectrophotometer (BioTek Instruments Inc., Winooski, VT, USA).

### 2.5. Colorimetric Assay for Chloride Quantification

Intracellular levels of chloride were quantified using Colorimetric Assay Kits (BioVision, Inc., Milpitas, CA, USA) with detection sensitivity of ~0.4 mM chloride, in accordance with the protocol provided by the manufacturer, using previously optimized amounts of whole-cell lysates and assay reagents as reported previously [22]. In the assay, mercuric ions combine with chloride ions to produce mercuric chlorides. The mercuric chloride frees a color-quantifiable complex, iron-TPTZ (2,4,6-Tripyridyl-S-triazine), which is measured by absorbance at 620 nm on a spectrophotometer to estimate chloride concentration. After concentrations were measured, the measurements were normalized to background, and results were calculated in relation to the standard curve. Concentrations (C) of chloride were calculated based on the formula, Ay/Sv = C (mM), where Ay denotes the amount of chloride (nmol) in sample wells from standard curve; Sv is the sample volume (μL). Chloride levels were read from the linear graph (*y* = 257.6*x* + 0.0278; R^2^ = 0.9873) generated from the data.

### 2.6. Preparation of Animals

This protocol has been described elsewhere [20]. Briefly, near end-term pregnant Sprague Dawley rats (*n* = 5 per group) were acquired from Jackson Labs and maintained until parturition. Two hours after birth, offspring and mothers of the experimental group were housed for 1 h within a chamber in which oxygen concentration alternated between room air (approximately 21% O_2_) and hypoxic conditions (approximately 10% O_2_) every 4 min, as described previously. We treated the pups for 1 h because we found that 1 h treatment (or 7 cycles of IH exposure) (1) reflects a clinically common condition, (2) is sufficiently mild not to result in conspicuous neural or β-cell destruction. Offspring and mothers of the control group were exposed to ambient air. Litter size was matched between the control and IH groups to help equalize weights of pups. After IH treatment, pups were maintained in ambient air until euthanasia. Female pups were excluded from the study, since earlier studies found potential sex differences on tissue effects from IH [23]. All animals were euthanized when pups were 3 weeks old. All processes involved in the experiments followed the National Institutes of Health guide for the care and use of Laboratory animals as approved.

### 2.7. RNA Extraction and Quantitative Real-Time RT-PCR

Total RNA was extracted from the harvested islets using the RNeasy Mini Kit (QIAGEN Sciences). First-strand cDNA was synthesized from the same amounts of RNA using a High-Capacity cDNA Reverse Transcription Kit (Applied Biosystems) and a random primer mixture. From the reverse transcriptase product, 2 µL of cDNA template was amplified with SYBR green master solution (Applied Biosystems) and gene-specific primers (Table 1) using the Eppendorf Realplex System (Eppendorf North America, Enfield, CT, USA). PCR was performed in triplicate for each sample obtained from 3 different pups in each group. All measured threshold cycle (Ct) values were normalized to a β-actin control.

### 2.8. Western Blot Assays

Whole-cell lysates were prepared using RIPA lysis buffer containing protease inhibitor cocktail. Proteins were resolved using sodium dodecyl sulfate–polyacrylamide gel electrophoresis (SDS-PAGE), and transferred onto a polyvinylidene difluoride (PVDF) membrane. The membrane was blocked with 5% milk tris-buffered saline-Tween (TBS-T), and incubated with primary antibodies for NKCC1 and KCC2 (sc-21545 and sc-367096; Santa Cruz Biotechnology, Santa Cruz, CA, USA), followed by a horseradish peroxidase-conjugated secondary antibody. Immunoreactive proteins were detected on X-ray films using chemiluminescent reagents. Cytosolic and plasma membrane fractions were prepared as needed, using a subcellular protein fractionation kit (Pierce Biotechnology Inc., Rockford, IL, USA). Proteins (50 μg/sample) were resolved on membranes using SDS-PAGE, and were analyzed by Western blot. Primary antibodies for KCC2, NKCC1 and pan-Cadherin were used. The density of each protein band was imaged using Multi Gauge v3.0 (Fujifilm USA, Los Angeles, CA, USA), and normalized with respect to the density of pan-Cadherin, a house-keeping protein in the plasma membrane.

### 2.9. Ethical Approval

The experimental protocol was approved by the Institutional Animal Review Committee IACUC (#D121101). All experiments were performed in accordance with the relevant institutional guidelines and regulations.

### 2.10. Euthanasia and Tissue Procurement

One week or 3 weeks after birth, experimental and control pups were fasted for 2 to 3 h prior to euthanasia. For tissue procurement, the animals were sacrificed in a CO_2_ chamber, and blood was drawn from the left ventricle of the heart immediately. The pancreas was rapidly harvested from each animal, and stored at −80 °C. Blood and tissue samples were harvested, and serum was separated from centrifuged blood and secured for assays.

### 2.11. Statistics

Statistical analyses were performed with SPSS v. 21. All statistical tests were two-tailed; *p* < 0.05 was considered significant. For group comparisons between subgroups (control vs. IH), *t*-tests and one-way ANOVA were performed. Tissue samples from at least three animals or three independent cultures were measured 3 times for each quantitative PCR, ELISA assay, and colorimetric evaluation.

## 3. Results

### 3.1. IH Exposure to Primary Islets Elevates KCC2 Transporters Participating in the Regulation of Insulin Secretion

We postulated that a regulator determining hyperpolarization or depolarization of the β-cell membrane for insulin secretion could hinge on intracellular chloride levels, which are determined by the density of chloride transporters, KCC2 [14]. In order to perturb KCC2 levels, we applied a hypoxic gas mixture (1% O_2_, 5% CO_2_, 94% N_2_) and 21% O_2_ ambient air, alternatively every 10 min for ten cycles. We determined the effects of disrupted KCC2 function in isolated primary islets in vitro, based on the assumption that the results would be equivalent to the physiologic outcomes of IH challenge to pancreatic islets in rat pups in vivo. Each set of results was evaluated by comparisons between control conditions (C) vs. IH exposed (IH) conditions, and by comparisons between scrambled-control (si-C) vs. KCC2-knock down (si-KCC2).

The level of KCC2 mRNA decreased significantly in primary islets after RNA knockdown even when knockdown was carried out after exposure to IH (Figure 1a). The results indicated that KCC2-siRNA down-regulated KCC2 mRNA levels in both control and IH treated islets. The siRNA-mediated knockdown resulted in up-regulation of insulin mRNA in both control and IH-treated islet cells (Figure 1b). We compared insulin 1 mRNA levels only because we quantified insulin protein levels subsequently. The increase in insulin 1 mRNA levels in KCC2 knockdown islets compared to scrambled controls (si-C) was greater in IH cells than in control cells (Figure 1b). These findings suggest that KCC2 influences insulin mRNA production. The efficiency of knockdown of KCC2 was assessed by Western blot analysis. The results confirmed that the KCC2 protein was significantly reduced by the si-KCC2 knockdown strategy (Figure 1c). ELISA results showed that KCC2 knockdown cells released significantly more insulin than si-scramble-control cells in both control and IH islets (Figure 1d). Further, the amount of insulin secreted by IH-treated islets after KCC2 knockdown approximated the amount secreted by the control islets (Figure 1d). We found that levels of total intracellular chloride declined approximately 20% (*p* = 0.0005) in IH-exposed islets (Figure 1e, compare solid bars C vs. IH open bars). When KCC2 transporters were disrupted by KCC2 knockdown, insulin secretion (Figure 1d, *p* < 0.0006) and intracellular chloride (Figure 1e, *p* < 0.0015) returned to near baseline levels even after IH challenge. These results indicate that the IH-induced effect on chloride levels was negated by KCC2 knockdown.

### 3.2. KCC2 Inhibitor Elevates Insulin Secretion and Chloride Levels

Results of loss-of-function studies on KCC2 chloride exporters using the KCC2 antagonist (VU0240551, 50 µM) showed that insulin release from both control and IH-treated pancreatic islets was significantly increased (*p* < 0.0001) (Figure 2a). Yet, the secreted amounts of insulin after VU0240551 treatment were similar before and after IH exposure (Figure 2a). These observations are consistent with our hypothesis that the KCC2 transporter plays a significant role in insulin homeostasis *via* regulating intracellular chloride levels in pancreatic β-cells. When KCC2 transporters were inhibited by a KCC2 antagonist, intracellular chloride levels (Figure 2b, *p* = 0.0021), as well as insulin secretion (Figure 2a, *p* < 0.0001) were elevated. Western blots exhibited that the amount of KCC2 transporters in the islets was not changed significantly by the KCC2 inhibitor (Figure 2c).

### 3.3. IH-Exposure for 1 h to Rat Pups at Postnatal Day 1 Decreases NKCC1, Increases KCC2 Levels, and Reduces Chloride Levels in Pancreatic Islets 3 Weeks Post-Exposure

We tested effects of IH exposure on mRNA and protein levels of K_ATP_ channel-associated genes (Sur1 and Kir6.2) in comparison to KCC2 between 1 h ambient air-treated (C, solid bars) and 1 h IH-treated (IH, open bars) rat pups (Figure 3a). The mRNA and protein levels of KCC2 increased (*p* < 0.003); whereas, mRNA and protein levels of Sur1 and Kir6.2 showed no change in IH-treated animals (Figure 3a,b). Following IH exposure to the pups on postnatal day 0 (P0), KCC2 levels markedly increased in membrane and cytoplasm in the animals in 3 weeks of age (Figure 3c). At 3 weeks post IH exposure, a substantial increase in KCC2 levels in both plasma membrane and cytoplasm was observed (Figure 3c); whereas, the level of NKCC1 decreased, particularly in the membrane. We also observed that the relative intracellular chloride content declined significantly (*p* = 0.004) (Figure 3d).

## 4. Discussion

‘Sudden onset diabetic symptoms’ exhibited by patients with COVID-19 are perplexing, showing mixed characteristics of Type 1 and Type 2 diabetes mellitus [3,24]. Recent publications have assumed destruction of insulin-producing β-cells by the virus; however, no clear cause–effect relationship has been offered [4,25]. Ample clinical evidence indicates that a relationship between COVID-19 and diabetes exists [26]. A recent study on an Italian cohort reported that 46% of patients were hyperglycemic, whereas 27% were normoglycemic [2], and found altered glycometabolic control, with insulin resistance and an abnormal cytokine profile, even in normoglycemic patients. This ‘new-onset hyperglycemia’ was persistent approximately 6 months in 35% of cases, but diabetes mellitus was diagnosed only in approximately 2% after recovery [27]. Their observations appear to indicate that such hyperglycemia may result from temporarily disturbed β-cell function, not from β-cell destruction, or even from the trans-differentiated SARS-CoV-2 infected β-cells as suggested recently [28]. We suspect a functional interplay between the sudden-onset diabetes and defective breathing in COVID-19 patients with no personal or familial history of diabetes mellitus [3,24].

Several studies have reported the occurrence of hypoxia-induced diabetes. Brief IH exposure to neonatal rat pups at postnatal day 1–2 resulted in decreased levels of insulin and an increased level of glucose in serum 3 weeks later, in addition to disturbed glucose tolerance [8]. This abnormality continued for at least 1 year with this intervention at an early age in rats. Similar outcomes are corroborated by other reports [12]. Plausible mechanisms presumed to be a ‘culprit’ for the COVID-19-associated diabetes include the co-existing ‘cytokine storm’ [2], potential structural damage to the β-cells [4,25] or trans-differentiation in the β-cells due to viral infection [28]. However, our etio-pathophysiology based on the current results is suggestive of an imbalanced chloride level in the β-cells as associated with hypochloremia commonly shown in patients with COVID-19 [8,9,29]. It is also unclear whether the symptoms observed in the current study represent an enterovirus-induced Type-1 diabetes [30] or Type-2 diabetes resembling the ‘sudden-onset’ diabetes shown in some patients with SARS and COVID-19 [2,4,25]. Nevertheless, intracellular chloride levels of the β-cells in the patients with COVID-19 should be lower than that of normal levels; thus, we assert that the hypochloremia could exacerbate the sudden onset diabetes found in patients with COVID-19. Because low chloride levels in our rat model triggered a lack of insulin secretion (as observed by others as well) after a brief IH challenge, we expect to observe an increased chance of hypo-insulinemia or hyper-glycemia if hospitalized COVID-patients showing hypochloremia with breathing difficulty.

Here, we assessed the effects of intermittent exposure to hypoxic conditions on rat pups, specifically on pancreatic β-cells. We found that the plasma membrane of pancreatic islet cells of rat pups contains a functional chloride transporter, KCC2. Both mRNA and protein levels of the chloride exporter, KCC2, in pancreatic islet cells increased markedly after IH exposure. Finally, insulin production and secretion appeared to be influenced by chloride-extruding KCC2 transporters and intracellular chloride content. β-cell membranes depolarize in response to rising blood glucose concentrations when intracellular chloride is maintained above its threshold potential [13]. Importantly, postnatal days 0 to 2 are known to be a critical window for an abrupt change in electrical activity as well as ion transporters in the plasma membrane [31]. Although the underlying mechanisms remain to be determined, the ensuing subcellular damage in β-cells expressed as a lack of intracellular chloride could result in a relative reduction in insulin release. Thus, the elevated oxidative stress conditions experienced by patients with impaired ventilation or/and reduced numbers of oxygen delivering red blood cells [7] could lead to acute severe diabetic symptoms [4] irrespective of the K_ATP_-channels [13,14].

Our findings indicate that intracellular chloride levels could be influenced by periodic hypoxic exposure. A series of studies by Di Fulvio’s group demonstrated that chloride homeostasis must be maintained to maintain β-cell membrane depolarization for adequate insulin secretion [13]. Down-regulation of KCC2 exporters increases insulin production and secretion markedly. Our observations, as well as by others, are consistent with the hypothesis that chloride transporters influence insulin homeostasis irrespective of the well-known ATP-dependent potassium-channels and Glut2 densities in the β-cell membrane after IH exposure. Of further interest, we found that inhibition of KCC2 mRNA restored insulin production to control levels. This ‘rescue-potential’ following KCC2 downregulation was greater than KCC2 mRNA levels of normal controls. This finding suggests that pharmacologic targeting of KCC2 transcription might have therapeutic potential for such patients. Given that we observed a substantial increase in KCC2 protein levels in the membrane and the cytoplasm of islets harvested 3 weeks post IH exposure, it will be important to determine the mechanisms by which KCC2 transporter or intracellular chloride levels participate in insulin production and secretion in COVID-19 patients suffered by hypoxia.

Our observations on rat pups imply a potential relevance to diabetic sequelae frequently observed in the patients with COVID-19, suggesting a β-cell dysfunction by chloride imbalance. Indeed, a disturbance in chloride levels was evident as reported among some patients with COVID-19 [8,9,29] which could be an ongoing threat if occurring, particularly to children with SARS-CoV-2 infection. Although still hypothetical, considering the urgency, it would be critical to fully elucidate the effects of chloride importer and exporters in SARS-CoV-2-infected pancreatic β-cells as a more disease-targeted remedial tactic than controlling hyperglycemic levels.

## Figures and Tables

**Figure 1 children-08-01178-f001:**
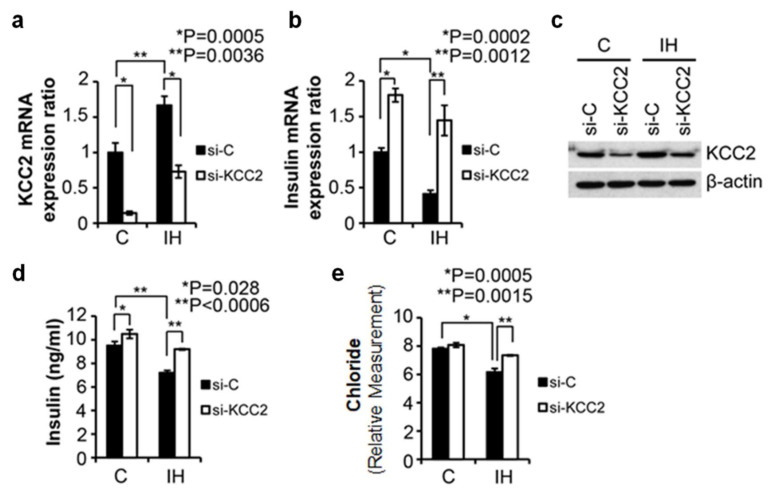
Assessing roles of KCC2 transporter in insulin production/secretion in isolated islets in vitro. (**a**). Relative KCC2 mRNA expression ratios measured in control (C) and IH-exposed islets (IH) treated with scrambled siRNA (si-C, solid bars) or KCC2 siRNA (si-KCC2, open bars). (**b**). Relative insulin 1 mRNA expression ratios measured in control and IH-exposed islets treated with scrambled siRNA (Si-C) or KCC2 siRNA (Si-KCC2). (**c**). Western blot analysis of KCC2 in control and IH-exposed islets treated with scrambled siRNA or KCC2 siRNA. Please find full-length blots for the cropped images in the Appendix A. (**d**). Insulin protein levels secreted into the medium measured by ELISA in control and IH-exposed islets treated with scrambled siRNA or KCC2 siRNA. (**e**). Colorimetric assay of relative chloride levels measured in control and IH-exposed islets treated with scrambled siRNA or KCC2 siRNA. Error bars indicate Standard Errors (S.E.) throughout figures. * *p* < 0.05, ** *p* < 0.01.

**Figure 2 children-08-01178-f002:**
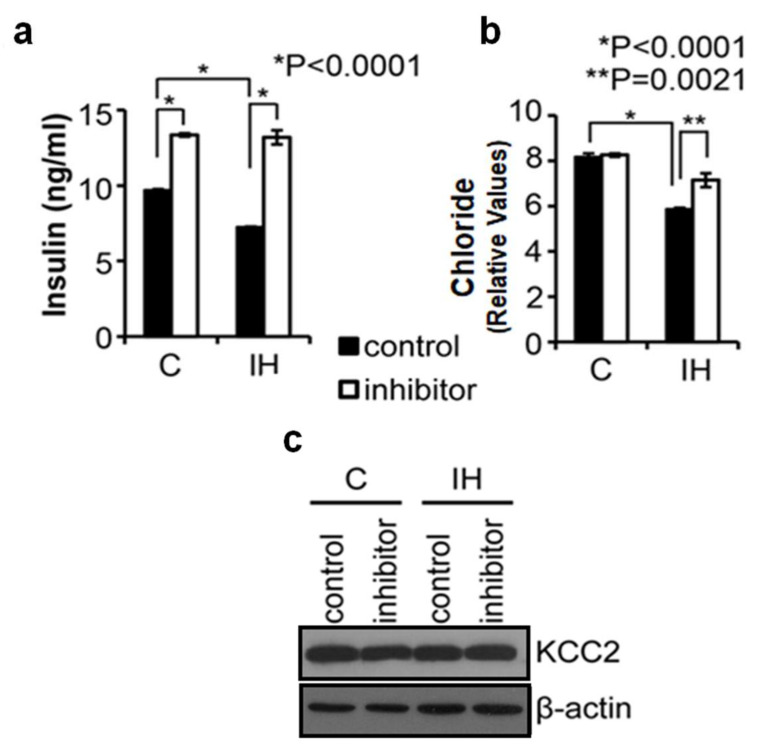
KCC2 inhibitor elevates insulin secretion as the level of intracellular chloride elevates in both control and IH-exposed islets in vitro. Control islets (solid bars) were treated with vehicle, whereas inhibitor islets (open bars) were treated with KCC2 antagonist. (**a**). Levels of insulin released from control and IH-exposed islets after KCC2 inhibitor treatment. (**b**). Colorimetric assay of relative chloride levels in control and IH-exposed islets treated with KCC2 inhibitor or vehicle (control). (**c**). Western blot analysis of KCC2 transporter levels in control and IH-exposed islets treated with inhibitor or vehicle. Please find full-length blots for the cropped images in the Appendix A. * *p* < 0.05, ** *p* < 0.01.

**Figure 3 children-08-01178-f003:**
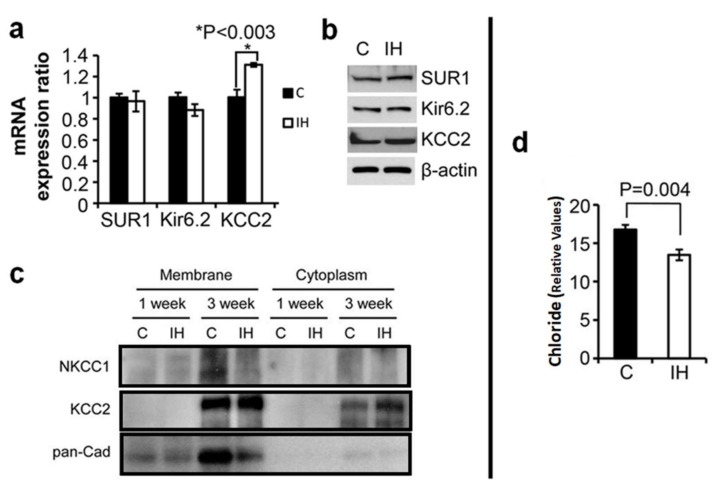
An increase in KCC2 transporter and a decrease in chloride levels in pancreatic islets after IH challenge in vivo compared to control. (**a**). mRNA level changes in Sur1, Kir6.2, and KCC2 3 weeks after IH exposure compared to control pups. (**b**). Western blots showing protein levels for KCC2 3 weeks after IH exposure to pups compared to control. Full-length blots for the cropped images are in the Appendix A. Note that levels for Sur1 and Kir6.2 showed no significant change, while KCC2 expression levels were elevated after IH exposure. (**c**). Western blot analysis of membrane and cytoplasmic protein levels of NKCC1 and KCC2 in pancreatic tissue measured 1 and 3 weeks after IH exposure compared to controls. Pan-Cadherin (pan-Cad) level indicates a reference protein in plasma membrane. (**d**). Relative chloride levels in cells harvested from IH and control pups at 3 weeks. Please find full-length blots for the cropped images in the Appendix A. * *p* < 0.05.

**Table 1 children-08-01178-t001:** Sequences of gene-specific primers.

Gene		Sequence (5′→3′)	
SUR1	NM_013039	F: TGCCTATGTCTTGGCTGTTC	R: CTCTCAGGTTGATCCCAGTTTC
Kir6.2	NM_031358	F: AGCCCAAGTTTAGCATCTCTC	R: GCACTCTACATACCGTACTTCAC
KCC2	NM_134363	F: TCCACCCAATTTCCCGATTT	R: CGTGTGGTCACTGTCTCATT
Ins1	NM_019129	F: ATCTTCAGACCTTGGCACTG	R: GGCTTTATTCATTGCAGAGGG
β-actin	NM_031144	F: ACAGGATGCAGAAGGAGATTAC	R: ACAGTGAGGCCAGGATAGA

## Data Availability

There is no other data supporting our reported results.

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
