# Peer review of "Potential Mechanisms Underlying Hypoxia-Induced Diabetes in a Rodent Model: Implications for COVID-19"

_children, 2021, doi:10.3390/children8121178_

Round 1

Reviewer 1 Report

Congratulations

Dear Sir/Mam
Please find bellow the requested review regarding the manuscript. The article contains a lot of useful information on the issue. The topic is very interesting and use of sources is appropriate. In addition, tables are very useful although some of them contain too much information. 
The article contains a lot of useful information on the issue. It is quite clear what is already known about this topic and the research question is clearly outlined. The abstract is too brief and Discussion section involve too much information. There must be a balance in the manuscript
Specifically
Discussion 
Discussion section is too long, while introduction section doesn’t involve too much information. There is an asymmetry in the manuscript.
Positive: There are some strengths of the article that could have an impact in the field, such as the topic and its impact on the existed literature. 

Author Response

The authors express sincere appreciations to the Reviewer 1. In response to the appropriate critiques, we have improved briefness of the abstract by adding a sentence for a smooth induction. We also deleted one entire paragraph from the discussion section. We believe these changes satisfies the Reviewer 1.

Reviewer 2 Report

     Minor points:

  • authors may elaborate a little more on the why for 1 h within a chamber in which oxygen concentration alternated between room air (approximately 21% O2) and hypoxic conditions (approximately 10% 137 O2) every 4 min and why they chose these time points.
  • In line 297, the clarity of the presentation needs to be improved.
  • A graphical abstract at the end of the Discussion may further make the paper more reader-friendly.

Author Response

The authors want to express sincere appreciations to the Reviewer 2. As shown in the Tract Change version, the reason for 1 h hypoxic challenge was elaborated. Line 297 has been revised now. We agree that a graphical abstract could help understand the contents. However, adding one more figure for the summary of verbal elaboration may be redundant.